# Pipeline Parallelism with Controllable Memory

**Penghui Qi**[*12], **Xinyi Wan**[*1], **Nyamdavaa Amar**[†2], **Min Lin**[1]
[1]Sea AI Lab    [2]National University of Singapore
{qiph,wanxy,linmin}@sea.com amara@u.nus.edu

## Abstract

Pipeline parallelism has been widely explored, but most existing schedules lack a systematic methodology. In this paper, we propose a framework to decompose pipeline schedules as repeating a building block, and show that the lifespan of the building block decides the peak activation memory of the pipeline schedule. Guided by the observations, we find that almost all existing pipeline schedules, to the best of our knowledge, are memory inefficient. To address this, we introduce a family of memory efficient building blocks with controllable activation memory, which can reduce the peak activation memory to 1/2 of 1F1B without sacrificing efficiency, and even to 1/3 with comparable throughput. We can also achieve almost zero pipeline bubbles while maintaining the same activation memory as 1F1B. Our evaluations demonstrate that in pure pipeline parallelism settings, our methods outperform 1F1B by from 7% to 55% in terms of throughput. When employing a grid search over hybrid parallelism hyperparameters in practical scenarios, our methods demonstrate a 16% throughput improvement over the 1F1B baseline for large language models. The implementation is open-sourced at this url.

## 1 Introduction

Distributed model training has attracted a lot of attention in recent years, especially after the boom of large language models [Brown et al., 2020]. As the model size becomes larger and larger, data parallelism (DP) [Goyal et al., 2017] is no longer capable to hold all the parameters in a single device. Under this background, model parallelism [Harlap et al., 2018, Huang et al., 2019, Shoeybi et al., 2019, Zheng et al., 2022] is proposed to partition parameters into a set of devices to address the memory constraint. Tensor parallelism (TP) [Shoeybi et al., 2019] is a commonly used model parallel strategy, which partitions weight parameters into several devices and performs matrix multiplication separately. A well-known shortcoming of TP is that, it requires a lot of communication volume, which makes it inefficient when bandwidth becomes the bottleneck Narayanan et al. [2021]. In such situations, pipeline parallelism [Harlap et al., 2018, Huang et al., 2019], which is another model parallel strategy, shows its advantage in low communication cost. The core idea of pipeline parallelism is to split the entire model into several stages, which can be processed by several devices in a streaming way. In a typical large-scale training scenarios such as Narayanan et al. [2021], TP is generally used within one compute node, and PP is used to scale up across nodes.

Although PP has been widely adopted and developed, it suffers from two prominent disadvantages: pipeline bubbles and large activation memory. To eliminate pipeline bubbles, one line of work focuses on asynchronous PP [Gaunt et al., 2017, Yang et al., 2021], which is theoretically bubble free. However, it sacrifices the exact optimization semantics and may result in lower convergence performance [Lian et al., 2018, Tang et al., 2020]. A parallel line of works revolve around synchronous PP, focusing on reducing pipeline bubbles and/or activation memory. GPipe [Huang et al., 2019] is

---

[*]Equal Contributors.

[†]Work was done during an internship at Sea AI Lab.

38th Conference on Neural Information Processing Systems (NeurIPS 2024).

an early work to reduce the bubble rate by increasing the number of microbatches, at the cost of more activation memory. 1F1B [Fan et al., 2021] avoids the activation memory growth with respect to the number of microbatches by staggering forward pass and backward pass, keeping the same bubble rate with GPipe. Another notable work is GEMS [Jain et al., 2020], which stores activation memory of only one forward pass by scheduling microbatches one after another among two model replicas, thus with a significantly large bubble rate. Chimera [Li and Hoefler, 2021] extends the ideas of GEMS by combining two pipelines in different directions together, which reduces pipeline bubbles when the number of microbatches is small, but with doubled parameter memory. Hanayo [Liu et al., 2023] is introduced to attain the same scheduling efficiency with Chimera without replicated models, but still suffering from scaling to more microbatches. Although its wave-like scheme is kind of similar to our V-shape building blocks, it is not motivated for memory balance, thus resulting in totally different pipeline schedules. In Megatron-LM [Narayanan et al., 2021], an interleaved strategy is proposed to further reduce the bubble rate, at the cost of more communication cost and a portion of extra activation memory. BPipe [Kim et al., 2023] focuses on reducing the activation memory of 1F1B from another perspective, transferring activations across devices based on the memory imbalance of 1F1B. However, it introduces a lot of extra communication and increases the complexity of the system, which makes it inefficient especially in settings with limited bandwidth. Zero Bubble [Qi et al., 2023] splits the backward into activation gradient computation and weight gradient computation, which can either reduce the pipeline bubbles without changing the maximum peak activation memory, or achieve zero bubble at the cost of doubled activation memory compared to 1F1B.

In this paper, we first demonstrate all existing pipelines can be seen as repeating a basic building block in time. We then identify a direct link between the activation memory and the lifespan of each building block, which reveals the core insight of this paper: lifespan decides the activation memory. Based on this insight, we present a family of novel and memory-efficient building blocks and their pipelines. Compared to 1F1B, we reduce the activation memory to 1/2 asymptotically with even higher throughput, and to 1/3 asymptotically with comparable throughput. We can also achieve zero bubble under the same activation memory with 1F1B. Notably, our strategy is almost a pure gain to the existing methods, only at the cost of doubled communication cost between pipeline stages, which is relatively small and can be neglected.

## 2 How to Build a Pipeline

We propose a four-step framework to design pipeline schedules.

**Building Block:** It starts by laying out the passes for a single microbatch, which we call a *building block*. For example, the building block of 1F1B is made of a sequence of forward passes followed by backward passes in the reverse order. We highlight the building block of 1F1B in color in Figure 1a.

**Repeating:** More microbatches are then introduced. The building blocks are repeated and woven together to form a pipeline. In Figure 1 (top), the repeating building blocks are shown in different shades of gray. Notably, legit building blocks are required to repeat without a collision, namely, the passes from two building blocks should not overlap with each other.

**Squeezing:** Depending on the building block, there may be redundant bubbles in the pipeline, which can be simply removed by squeezing without changing the order of the passes. For example, Figure 1b shows a case where squeezing produces a more efficient pipeline.

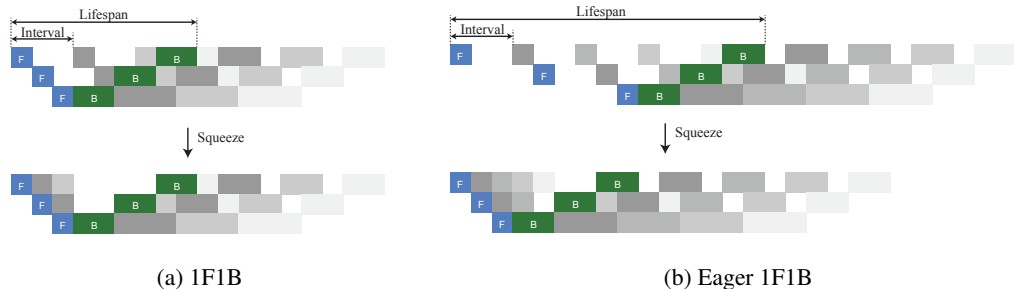

(a) 1F1B                                    (b) Eager 1F1B

Figure 1: A pipeline can be built by repeating a building block, and then squeezing redundant bubbles.

**Reordering (optional):** We can reorder the passes in the warm-up and cool-down phase to further improve the computation throughput. Intuitively, the peak of memory happens in the stable phase of the pipeline, while in the warm-up and cool-down phases the RAM is under utilized, leaving some space for improving the computation throughput without changing peak memory. We leave the details in Appendix C.

Most of existing pipeline schedules can be explained under this framework. Besides the 1F1B and eager 1F1B shown in Figure 1, we show the interleaved 1F1B [Shoeybi et al., 2019], ZB-H1 [Qi et al., 2023] and a series of well-known pipelines in a more extensive gallery (see Appendix I).

## 2.1 Building Blocks

The computation and memory efficiency of various pipelines can be attributed to their building blocks. The diversity of the building blocks primarily comes from three factors, **model partitioning**, **device placement**, and **offsets** between passes. We follow the idea and notations in zero bubble PP [Qi et al., 2023], using $F$ to denote forward pass, $B$ to denote "backward for the activations", and $W$ to denote "backward for the weights". Note that such finer granularity can be generalized to previous methods like 1F1B, by always grouping $B$ and $W$ together.

Model partitioning deals with how the model is divided into pipeline stages. The most common pattern is to equally divide the model to match the number of devices. A prominent example is the 1F1B schedule (Figure 1a). This is extended in interleaved 1F1B where the number of stages can be an integer multiple of the number of devices.

Device placement is another key factor in the design of building blocks. While conventionally each pipeline stage is sequentially placed on a different device, it is not uncommon to place multiple stages on the same device like in interleaved 1F1B (Figure 18h). Another example of unconventional device placement is Chimera, where two pipelines are placed in reversed device order.

Last but not least, the offsets between $F,B,W$ passes play a major role in the computation and memory efficiencies of the pipeline. By simply enlarging the offsets between subsequent $F$ passes in the building block of 1F1B, we obtain the eager 1F1B [Zhuang et al., 2023] (Figure 1b) where more $F$ passes are eagerly scheduled, resulting in higher memory consumption (but better communication overlapping). GPipe can be seen as adding a large offset between the last $F$ and the first $B$ in the 1F1B building block. One more example on the effect of the offset is the comparison of ZB-H1 (Figure 18c) and ZB-H2 (Figure 18d) schedules, one can see that properly chosen offsets result in zero bubble schedules like ZB-H2. In this work, we assume that every $F$, $B$ or $W$ pass takes equally one unit of computation time, and only consider integer unit of offsets. Although this may limit the number of feasible building blocks, it greatly improves the simplicity of analysis.

## 2.2 Calculating the Peak Memory

Not every pipeline is born equal, researchers are constantly looking for pipelines that are more efficient in computation and/or memory. While efficient pipelines could be discovered by enumerating every possible building block, it is nonetheless prohibitively expensive. We discover that the peak memory consumption of a pipeline can be calculated from its building block via a simple formula. This enables us to design pipelines with a controllable peak memory.

Two quantities are crucial for the calculation of peak memory, the *lifespan* of a stage, and the repeating *interval* of the building blocks, both of which are illustrated in Figure 1. The lifespan of a stage is defined as the amount of time between the starting of the $F$ pass and the ending of $B$ or $W$ pass. A piece of activation memory is allocated at the starting of $F$, and retained in the RAM throughout the lifespan until it is consumed by both $B$ and $W$. The peak memory consumption can be calculated by finding the maximum number of microbatches whose lifespans overlap with that of every other microbatch. Using $l$ to denote lifespan, $T$ to denote the repeating interval and $m$ the size of activation memory for a single microbatch, we have the relation.

$$\text{peak memory} \leq \lceil \frac{l}{T} \rceil m$$

When there are multiple stages on one device, e.g. interleaved 1F1B, their contributions to the peak memory are independent, using $S_i$ to denote all the stages allocated to device $i$, we sum the

contributions from every stage.

$$\text{peak memory of device } i \leq \sum_{s \in S_i} \lceil \frac{l^s}{T} \rceil m^s \qquad (1)$$

Another key insight is that the repeating interval $T$ is readily determined from the building block. In an efficient pipeline, $T$ should be equal to the number of units of computation in each stage of the building block. Any $T$ larger than that would cause pipeline bubbles in the stable phase, and $T$ smaller than that would lead to collisions. A subtle exception is the interleaved 1F1B whose repeating interval is not uniform. We leave the discussion to Appendix G.

## 2.3 Repeating without Collision

One constraint to keep in mind when designing the building blocks is that a legit building block is required to repeat without any collision. It may seem unintuitive how to design building blocks with this constraint. In practice, we design the building block first and perform a post-hoc verification. Another useful observation is that a legit building block usually produces a stable phase in the middle of the pipeline, which contains a repeating $d \times T$ rectangle, where $d$ is the number of devices and $T$ is the repeating interval. This offers an alternative to constrain the building blocks. We can start by ordering passes within this rectangle and convert it back to a building block.

## 3 Memory Efficient Building Blocks

With the above framework, we can conveniently analyze the memory consumption pattern of existing pipelines. To our knowledge, all existing pipelines are memory inefficient due to two primary reasons: redundant dependency chain, and imbalanced memory usage. Before Zero Bubble [Qi et al., 2023], the backward is often regarded as a single pass, resulting in unnecessarily longer lifespan thus more memory footprint. In this paper, we leverage the backward splitting strategy to remove these redundant lifespan. The imbalanced memory comes from the innate heterogeneity of the lifespans across stages. From Figure 1a, we can easily see that the lifespan of the stages differs greatly from each other, with the first stage having the longest lifespan. Consequently, it causes a memory bottleneck on the first device and under utilization of memory on all other devices. To resolve this problem, we introduce a family of novel building blocks, which we refer to as *V-Shape* building blocks. The core insight comes from Equation 1 which says that the peak memory depends on the sum of the lifespans. Therefore, when we place multiple stages on the same device, we should always collocate stages of long lifespans with those of short lifespans. When the total sum of lifespans is fixed, balanced placement always means higher memory efficiency. This can be demonstrated by Figure 2, the parallel schedule (used in interleaved 1F1B) is imbalanced and has a memory bottleneck proportional to $l_1 + l_4$, while in the V-Shape schedule it is $l_1 + l_6$.

The V-Shape schedule requests the model to be **partitioned** into stages twice the number of devices and the **device placement** of the second half of stages to be in reverse order as the first half. As the offsets directly determine the lifespan of each stage and therefore the peak memory by Equation 1, we can then further control the **offsets** between passes to generate building blocks with diverse memory.

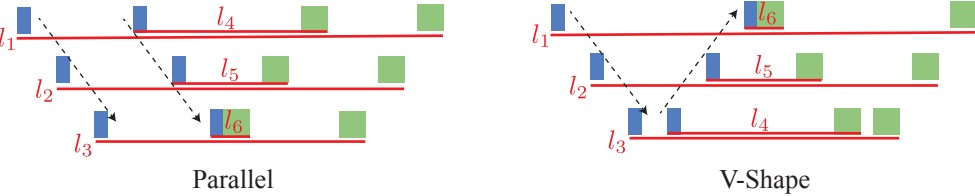

Figure 2: The V-Shape building block ensures balanced peak memory across all devices, whereas the parallel building block has a memory bottleneck in the first device.

## 3.1 Controllable Balanced Memory

We assume the model is uniformly partitioned, namely, both the computation and memory of each stage are identical. For a single microbatch, we denote the activation memory of each stage as $m$, and

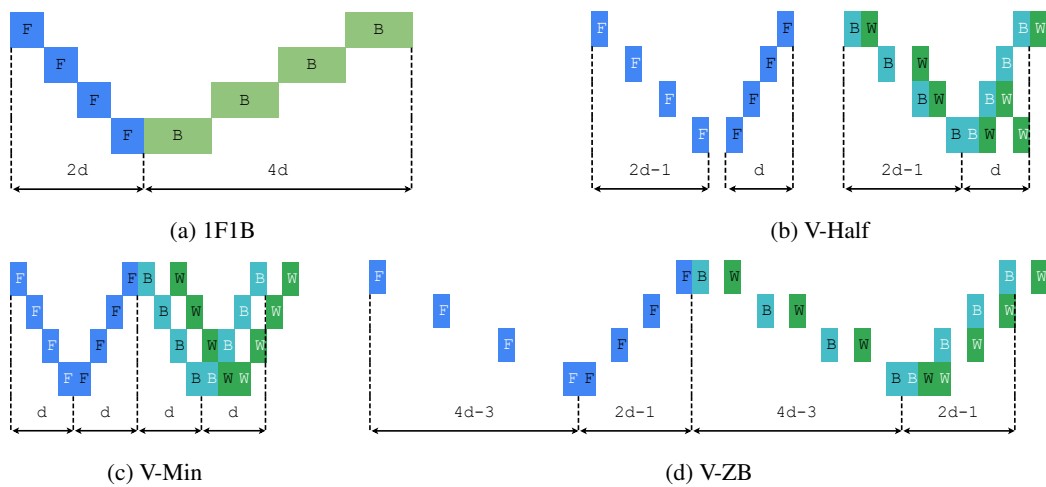

Figure 3: V-Shape building blocks with 4 devices ($d = 4$), where white text colors represent the first half of model stages and black text colors represent the second half. $F$, $B$ and $W$ represent the forward, backward (for activation gradients) and backward for weight gradients, respectively.

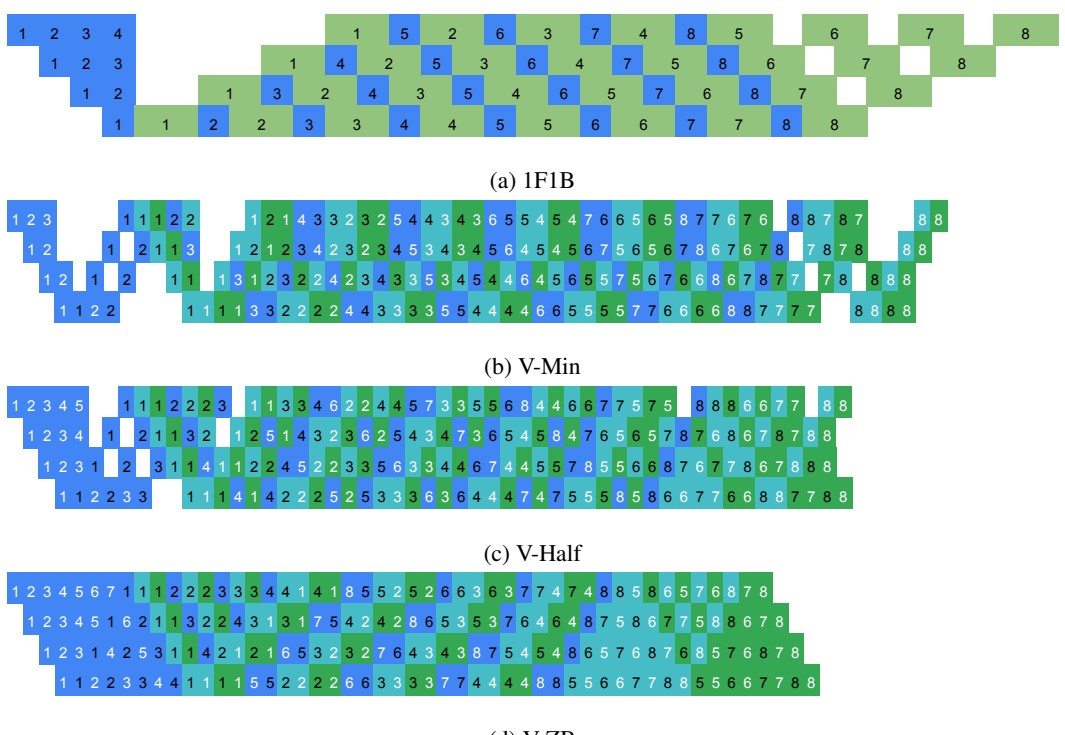

Figure 4: V-Shape schedules compared to 1F1B, under the setting of 4 devices and 8 microbatches. The stable phases adhere to the pattern of their building blocks.

the total activation memory of the entire model as $M$. Note that $M = 2dm$, where $d$ is the number of devices. To make it simple and tractable, we use **uniform offsets within each half of $F$ and $B$ passes** to control the peak memory. Specifically, we apply the same offset $\delta_F^0$ between two adjacent $F$ passes within the first $d$ stages (e.g., $\delta_F^0 = 2$ in Figure 3b, $\delta_F^0 = 1$ in Figure 3c and $\delta_F^0 = 4$ in Figure 3d). Similar constraints are applied to the other half of the $F$ passes and both halves of the $B$ passes, denoted as $\delta_F^1, \delta_B^0, \delta_B^1$, respectively. To guarantee balanced peak memory across devices, we add another two constraints, $\delta_F^0 = \delta_B^1 = \delta^0$ and $\delta_F^1 = \delta_B^0 = \delta^1$, where we use notations $\delta^0$ and

$\delta^1$ for simplicity. For example, in Figure 3d, we set $\delta^0 = 4$ and $\delta^1 = 2$. Note that we only control the offsets across different devices. For those adjacent passes within the same device (e.g., *F* and *B* of the last stage, two *F* and two *B* in the last device), we use brute force to find optimal solutions, ensuring their offsets are small (less than the repeating interval). Note that *W* can always be placed greedily after settling all *F* and *B* passes, so we don't need to search their offsets during brute force. According to Equation 1, we can analyze the asymptotic peak memory with respect to $d$,

$$\text{peak memory of device } i \leq \frac{2d(\delta^0 + \delta^1) + O(1)}{6} m \approx \frac{\delta^0 + \delta^1}{6} M \tag{2}$$

By ignoring the small constant, we can directly control the peak memory by the value of $\delta^0$ and $\delta^1$.

Table 1: Small constant values are ignored for bubbles and peak memory of *V-Min* and *V-Half*. For 1F1B, $\delta^0/\delta^1$ are redefined as the offsets between adjacent forward/backward passes. $M$ represents the total activation memory of the entire model, and $d$ is the number of devices.

| Building Block | $\delta^0$ | $\delta^1$ | Peak Memory | Bubbles |
|---|---|---|---|---|
| 1F1B | 2 | 4 | $M$ | $\approx 6d$ |
| *V-Min* | 1 | 1 | $\approx M/3$ | $\approx 4d$ |
| *V-Half* | 2 | 1 | $\approx M/2$ | $\approx 3d$ |
| *V-ZB* | 4 | 2 | $M$ | 0 |

### 3.2 V-Shape Pipeline Schedules

By varying the values of $\delta^0$ and $\delta^1$, we come up with 3 novel V-Shape building blocks (Figure 3), and present their final schedules based on our framework in Figure 4. The building block of *V-Min* (Figure 3c) has the minimum offsets, namely $\delta^0 = \delta^1 = 1$, thus the minimum memory consumption. With $\delta^0 = 4$ and $\delta^1 = 2$ as in Figure 3d, *V-ZB* eliminates the bubble to almost 0 (Figure 4d), pushing to extreme throughput. The building block of *V-Half* (Figure 3b), which uses $\delta^0 = 2$ and $\delta^1 = 1$, sits between the two extremes and consumes about half of the activation memory required by 1F1B. Although both *V-Min* and *V-Half* have lower memory footprint than 1F1B, *V-Min* contains about 2/3 and *V-Half* contains about 1/2 of 1F1B's bubbles, assuming *F*, *B*, *W* have equal run time. We show the comparison between our proposed V-Shape schedules and 1F1B in Table 1. Notably, the exact peak memory is $\lceil \frac{d+2}{3} \rceil \frac{M}{d}$ for *V-Min*, and $\lceil \frac{d+1}{2} \rceil \frac{M}{d}$ for *V-Half*. To avoid collisions in the building blocks of *V-Min* and *V-Half*, the offsets (within the same device) are slightly different for different values of $d$. The details are in Appendix F.

### 3.3 Repeating Bubbles in *V-Min*

In real-world scenarios where *F*, *B* and *W* have different run times, *V-Min* suffers from a repeating bubble. As shown in Figure 5, there exists bubbles for every repeating interval $T$. Consequently, the bubble grows as the number of microbatches increases. Although *V-Half* may encounter the same issue (when the times of *F*, *B* and *W* differ significantly), it generates patterns that tessellate well in most empirical cases due to its loose dependencies. As illustrated in Figure 5b, the throughput of *V-Half* is robust to the variation of run times. Additionally, the bubbles of *V-ZB* will never grow when increasing the number of microbatches. We leave the related discussions in Appendix E.

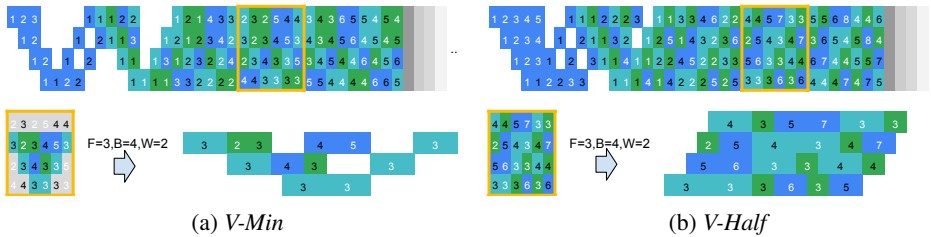

(a) *V-Min*            (b) *V-Half*

Figure 5: We take a repeating $d \times T$ grid from *V-Min* and *V-Half* schedules, and assign *F/B/W* with different values. The result shows *V-Min* has bubbles for every repeating grid, while *V-Half* does not.

### 3.4 Other Building Blocks

Besides V-Shape building blocks, we also propose some other interesting building blocks in Appendix H, to show the generalization ability of our framework. Some useful examples include a) 1F1B-V achieving 2/3 of 1F1B's activation memory without doing B-W split; b) **a schedule consumes less memory than interleaved 1F1B but with the same bubble rate (Figure 17c)**. Additionally, we design an adaptive scheduler to control the memory at a finer granularity in Appendix A.

## 4 Experiments

We construct our experiments to show three conclusions: a) The throughput and memory of *V-Min*, *V-Half* and *V-ZB* aligns with the theoretical analysis in Section 3.2; b) Memory-saving methods including *V-Min* and *V-Half* can bring accelerations; c) Our methods still perform best when combining with other state-of-the-art techniques.

### 4.1 Setup

We evaluate our methods using a series of models detailed in Table 2 analogous to GPT-3 [Brown et al., 2020]. Our implementation is based on the open-source Megatron-LM project [Narayanan et al., 2021] and is experimented on up to 40 NVIDIA A100 SXM 80G GPUs distributed across 5 nodes interconnected by a RoCE RDMA network. The running time of each iteration is recorded after several warm-up iterations. Similar to the settings in [Qi et al., 2023], we deduct one transformer layer from both the initial and final pipeline stage to compensate for the embedding and output layer in LM, which can otherwise become the bottleneck of the pipeline and interfere to the efficiency.

Table 2: Models used in experiments.

| Model | Layers | Attention Heads | Hidden Size | GPUs |
|-------|--------|-----------------|-------------|------|
| 9.6B | 30 | 40 | 5120 | 16 |
| 21B | 46 | 48 | 6144 | 24 |
| 38.5B | 62 | 64 | 7168 | 32 |
| 98.5B | 78 | 80 | 10240 | 40 |

Our experiments majorly focuses on the following pipeline parallel schedules: a) *V-Min*, *V-Half* and *V-ZB*: schedules introduced in Section 3.2; b) 1F1B and Interleaved 1F1B: methods implemented in Megatron-LM; c) 1F1B-R: 1F1B with full activation rematerialization [Chen et al., 2016]; d) ZB-1P and ZB-2P: the adaptive zero-bubble methods introduced in [Qi et al., 2023] with activation memory limit set to the 1x/2x times of 1F1B.

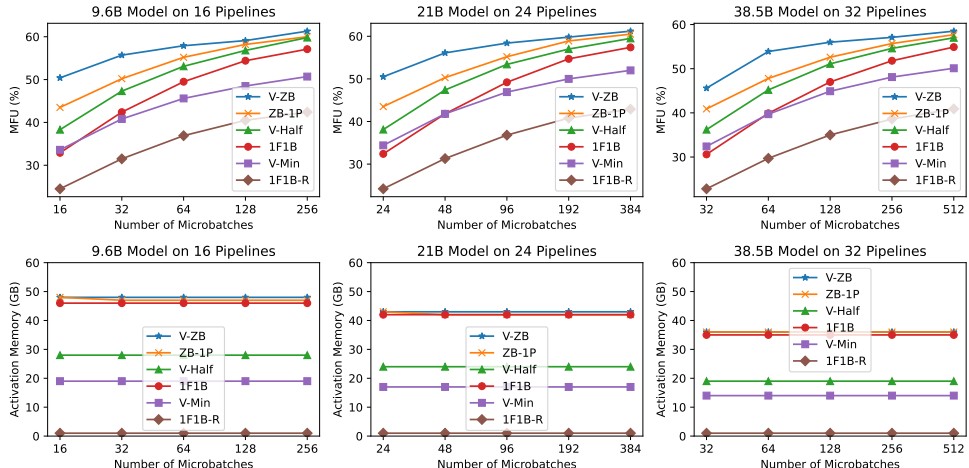

Figure 6: Throughput and activation memory using the same microbatch size.

## 4.2 Comparing Pipeline Schedules

In Figure 6, we present comparisons of the throughput measured in FLOPS utilization (MFU) and activation memory consumption across different pipeline schedules under various settings. From the results, *V-ZB* outperforms all other methods in terms of throughput, which aligns with Figure 4. When comparing the activation memory consumption, *V-Min* and *V-Half* stand out by significantly reducing activation memory to approximately 1/3 and 1/2, while other methods' memory is similar except for 1F1B-R. More details of our experiments and definition of metrics can be found in Appendix D.1.

Notably *V-Min* has a comparable throughput against 1F1B, but its throughput falls behind 1F1B at a larger number of microbatches due to the aforementioned repeating bubble in Figure 5a, as discussed in Section 3.3. However, it still outperforms 1F1B with full activation rematerialization, providing a strong alternative for saving memory.

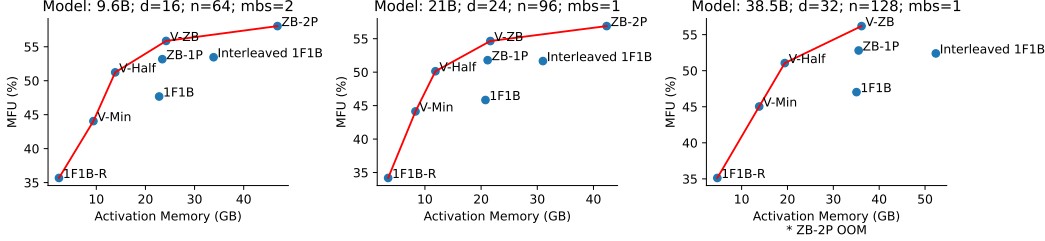

Figure 7: Pareto frontier of MFU and memory for various setups.

We also plot both memory and MFU for the various methods in Figure 7 in a typical, but slightly different setting in which we reduced the microbatch size of 9.6B and 21B model to allow ZB-2P and Interleaved 1F1B to run which would otherwise run out of memory (OOM). It shows that the V-Shape pipeline schedules lie at the Pareto frontier.

## 4.3 When to Save Memory

While *V-ZB* provides optimal throughput, *V-Half* and *V-Min* methods are mainly used when memory budget is tight. Conventionally, rematerialization is used when it runs out of memory (OOM). However, rematerialization leads to repeated computation and consequently decrease the throughput. *V-Half* and *V-Min* significantly outperforms rematerialization (1F1B-R) as we show in Table 7.

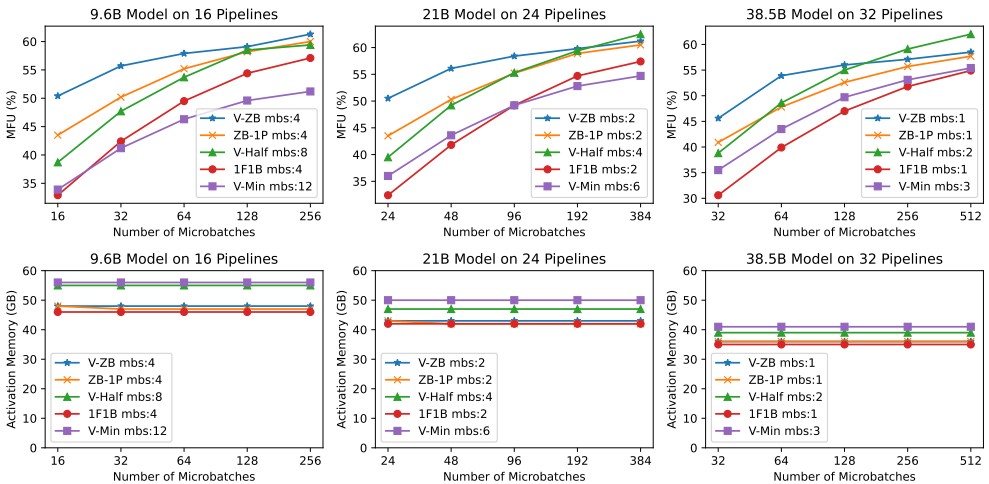

Figure 8: Throughput and activation memory under similar memory limit.

Another benefit of saving memory is that we can potentially use the extra memory for an increased microbatch size, which leads to a higher arithmetic intensity. We present the results in Figure

8. On bigger models, where memory pressure is higher and hence microbatch size is smaller, *V-Half* schedule can surpass *V-ZB* and other baselines because of its arithmetic intensity gain. This observation does not apply for *V-Min*, implying its arithmetic intensity gain can not compensate for the increased bubble. Doubling/Tripling the microbatch size for *V-Half*/*V-Min* results in a slightly higher activation memory than the other methods. This reflects the constant factors we ignored in Section 3.2. The increase is less significant as the number of devices grows.

## 4.4 Combining with Existing Techniques

We present our methods in the context of LLM training together with various other techniques. The following techniques are considered: a) Flash Attention [Dao et al., 2022, Dao, 2023]; b) Tensor Parallelism [Narayanan et al., 2021] and Sequence Parallelism [Korthikanti et al., 2023]; c) Distributed Optimizer provided in Megatron-LM. The implementations are all from Megatron-LM [Narayanan et al., 2021]. Both our methods and the baseline methods are combined with the above techniques. Similar to the evaluation method in Kim et al. [2023], we perform a grid search on the following parameters: the size of PP; the size of TP; the size of DP; the microbatch size ($mbs$). We use 40 GPUs in this experiment. For each method, the best result from the grid search is reported.

We present the best result for each pipeline parallel schedule in Table 3 and the corresponding parameters. We find that when sequence length is smaller and hence the memory budget is more abundant, *V-ZB* performs the best due to the elimination of bubbles. When we increase the memory pressure by increasing the sequence length, *V-Half* performs the best because of its memory efficiency. The detailed data and analysis of grid search can be found in the Appendix D.3 D.4.

Table 3: V-Shape schedules combined with other memory saving methods.

| Common Setup | PP Method | Best MFU (%) | Best Parameters | | | |
|---|---|---|---|---|---|---|
| | | | DP | TP | PP | $mbs$ |
| 98.5B Model Seq Length 1024 Batch Size 160 | 1F1B | 54.77 | 2 | 4 | 5 | 2 |
| | 1F1B-R | 40.84 | 2 | 2 | 10 | 1 |
| | ZB-1P | 59.95 | 1 | 1 | 40 | 1 |
| | V-Half | 57.83 | 2 | 1 | 20 | 1 |
| | V-Min | 52.8 | 2 | 2 | 10 | 1 |
| | V-ZB | **63.31** | 1 | 1 | 40 | 1 |
| 98.5B Model Seq Length 3072 Batch Size 640 | 1F1B | 62.95 | 2 | 4 | 5 | 1 |
| | 1F1B-R | 50.37 | 2 | 1 | 20 | 1 |
| | ZB-1P | 62.18 | 1 | 4 | 10 | 1 |
| | V-Half | **66.34** | 1 | 2 | 20 | 1 |
| | V-Min | 61.04 | 1 | 1 | 40 | 1 |
| | V-ZB | 62.56 | 1 | 4 | 10 | 1 |
| 98.5B Model Seq Length 16384 Batch Size 160 | 1F1B | OOM | - | | | |
| | 1F1B-R | 42.05 | 1 | 4 | 10 | 1 |
| | ZB-1P | OOM | - | | | |
| | V-Half | **57.85** | 1 | 8 | 5 | 1 |
| | V-Min | 48.58 | 1 | 8 | 5 | 1 |
| | V-ZB | OOM | - | | | |

## 5 Conclusion And Future Work

In this work, we present a framework that constructs pipeline schedules by focusing on their repeating building blocks. This framework enables direct computation of peak memory from the lifespan of the building block. Based on this capability, we design a family of memory-efficient building blocks. We discuss three representative methods from this family, namely *V-Min*, *V-Half* and *V-ZB*, and demonstrate with experiments that our methods advance the Pareto frontier of throughput and memory in large model training. Furthermore, our methodology of designing pipeline schedules through building blocks may inspire the research community to explore more novel pipeline schedules. Notice that repeating a building block is not the only way of building a pipeline, other methods like greedy search could generate a pipeline that has no repeating patterns.

In the future, we plan to further explore more memory efficient pipeline schedules based on our framework. A major limitation of *V-Min* is that, it suffers from growing bubbles when increasing the number of microbatches. Although *V-Half* mitigates this issue, there is still a space to further reduce the memory consumption. Using continuous offsets or finer-granularity discretization is a possible way to solve it. We leave it in our future work.

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

# A    Adaptive Scheduler Based on Search

Now we consider more general scenarios, where we want to minimize the bubbles given an activation memory limit. A straightforward approach should be simply searching over all possible offsets and picking the one with minimal bubbles. However, this naive method cannot work well due to there are exponentially many possible offsets, which makes it intractable to iterate thoroughly. In this section, we propose a more practical searching method to solve this general problem.

We use superscript $c \in \{0, 1\}$ to denote which stage in a device, and use subscript $i \in \{1, 2, ..., d\}$ to denote the index of the device. For example, $F_i^c$ represent the forward pass of stage $cd + i$ in the $i$-th device. We define the offset from $u$ to $v$ as $\delta(u, v) = t(v) - t(u)$, where $t(v)$ represent the cell index along time horizon of pass $v$. To simplify the notations, we define $\delta F_i^0 = \delta(F_i^0, F_{i+1}^0)$, $\delta F_i^1 = \delta(F_i^1, F_{i-1}^1)$, $\delta B_i^1 = \delta(B_i^1, B_{i+1}^1)$ and $\delta B_i^0 = \delta(B_i^0, B_{i-1}^0)$, to denote the offset from a pass to its next pass.

Instead of all possible offsets, we limit our search space to uniform offsets across devices. We also try to ensure each device has a balanced peak memory. Note that for the uniform offsets introduced in Section 3.1, the peak memory only falls into a small discrete set ($\{\frac{k}{6}M\}$, where $k$ is an integer). To make it work for a finer granularity of memory controlling, we split the repeating module into two parts, containing the first $K$ rows and the last $d - K$ rows respectively. More formally, we use the constraint as follows.

$$
\begin{aligned}
\delta F_i^0 = \delta B_i^1 = \delta_{<K}^0, \forall 1 \leq i < K \qquad \delta F_i^1 = \delta B_i^0 = \delta_{\leq K}^1, \forall 1 < i \leq K \\
\delta F_i^0 = \delta B_i^1 = \delta_{\geq K}^0, \forall K \leq i < d \qquad \delta F_i^1 = \delta B_i^0 = \delta_{>K}^1, \forall K < i \leq d
\end{aligned}
\tag{3}
$$

Note that the above constraints have good properties that the peak memory is balanced across devices. As we can always greedily fill $W_i^0$ and $W_i^1$ when repeating, we only need to search over the permutation of the first device, the values of $\delta_{<K}^0, \delta_{\leq K}^1, \delta_{\geq K}^0, \delta_{>K}^1$ and $K$. The computational complexity is $O(d)$ if we regard repeating interval as a constant.

For each building block searched, we repeat the building block, check collision, do squeezing and reordering as mentioned in 2.1. After searching over all possible building blocks, we pick the schedule with minimal bubbles. Note that we can use the true run times of $F$, $B$ and $W$ to calculate the bubbles, which will lead to more efficient schedule in real cases.

# B    Evaluation of Adaptive Scheduler

In Figure 9 we plot the bubble rate of adaptive V-Shape schedulers introduced in A under various settings and different memory limit. The run times of $F$, $B$ and $W$ are from profiled real cases, as in Table 5. We observe that the bubble rate drops as memory limit increases. Notably, there's a sudden drop in bubble rate when the memory limit just goes above approximately 1/2 of 1F1B, at which point the repeating bubble mentioned in Figure 5a disappears.

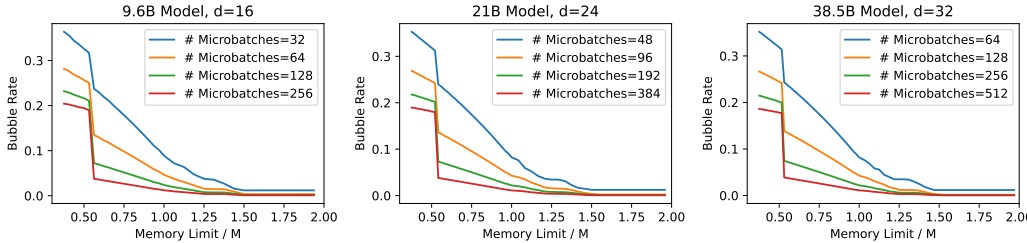

Figure 9: Bubble rate of V scheduler under various settings.

We also compare V scheduler with the adaptive zero bubble scheduler proposed in [Qi et al., 2023] in Figure 10. We find that V scheduler has a boarder range of memory configurations and a smaller bubble rate compared to zero bubble scheduler. We also draw the bubble rate of 1F1B as a reference, though 1F1B does not support a configurable memory.

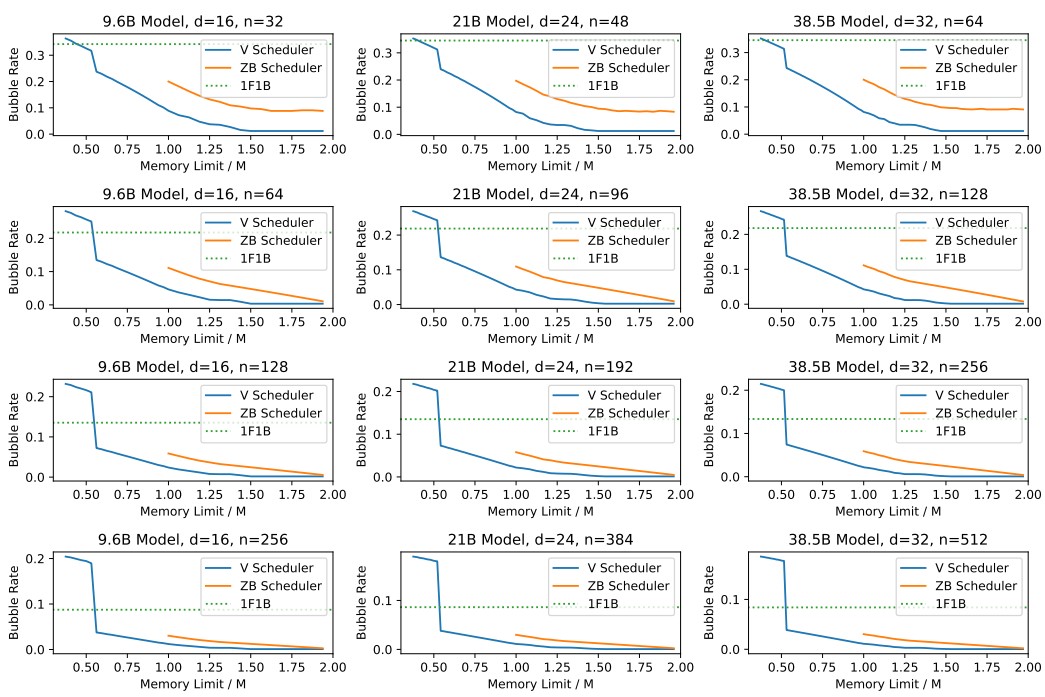

Figure 10: Comparison of V scheduler and zero bubble scheduler.

# C  Reordering

In our framework, we may need to reorder the warm-up and cool-down phase after squeezing. Basically, we employ simple greedy approaches to handle the reordering for warm-up and cool-down, and illustrate how zero bubble schedule is reordered in Figure 11.

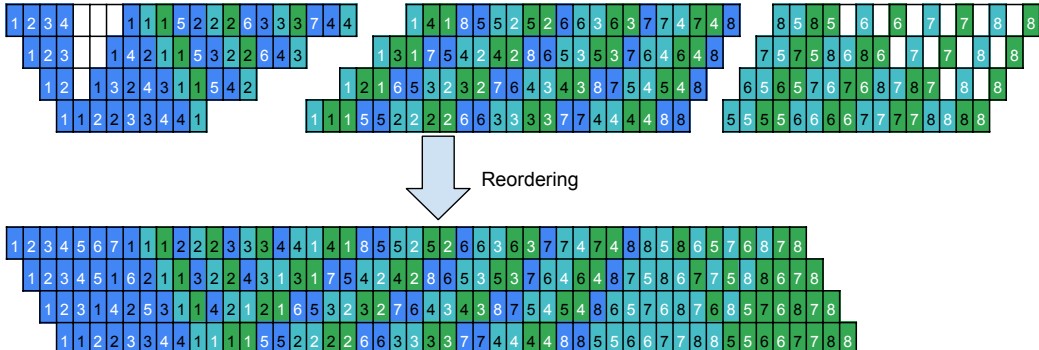

Reordering

Figure 11: Top: the schedule after **Squeezing**. Bottom: the schedule after **Reordering**.

**Warm-up**   In warm-up phase, bubbles mainly happen before the first $B$. We iterate all the cells from left to right. If a vacant cell (which means a bubble) is encountered, we try to find a computation pass to fill this bubble. We iterate all the following computation passes in the same device, and check whether it is possible to move if we keep all other passes unchanged. If the check succeeds, we move it to the vacant cell, and the bubble is filled.

**Cool-down**   In cool-down phase, $W$ can be scheduled at any time after its corresponding $B$. So we utilize a heuristic way to handle the bubbles. Firstly, we delete all the $W$ passes in cool-down phase. Next, we squeeze the schedule to remove the bubbles caused by deleting $W$. After that, we use $W$ to

fill the remaining bubbles, ensuring each $W$ is after its corresponding $B$. Finally, we schedule all the remaining $W$ passes at the end.

Despite its simplicity, the above heuristics is general and effective. However, it may not achieve the best performance in some cases. We also design other greedy or constructive methods as a complement for some building blocks. We will release all the related code in our repository.

# D    Detailed Experiment Data

Table 4: Comparing Pipeline Schedules

| Setup | Model | 9.6B | | | | | 21B | | | | | 38.5B | | | | |
|---|---|---|---|---|---|---|---|---|---|---|---|---|---|---|---|---|
| | #GPU | 16 | | | | | 24 | | | | | 32 | | | | |
| | Microbatch | 4 | | | | | 2 | | | | | 1 | | | | |
| | #Microbatch | 16 | 32 | 64 | 128 | 256 | 24 | 48 | 96 | 192 | 384 | 32 | 64 | 128 | 256 | 512 |
| Samples per second per GPU | V-ZB | 2.59 | 2.86 | 2.98 | 3.04 | 3.15 | 1.19 | 1.32 | 1.38 | 1.41 | 1.44 | 0.59 | 0.70 | 0.72 | 0.74 | 0.76 |
| | ZB-1P | 2.24 | 2.58 | 2.84 | 2.99 | 3.08 | 1.02 | 1.18 | 1.30 | 1.39 | 1.43 | 0.53 | 0.62 | 0.68 | 0.72 | 0.75 |
| | V-Half | 1.97 | 2.43 | 2.73 | 2.92 | 3.08 | 0.90 | 1.12 | 1.26 | 1.34 | 1.40 | 0.47 | 0.58 | 0.66 | 0.71 | 0.74 |
| | 1F1B | 1.69 | 2.18 | 2.55 | 2.80 | 2.94 | 0.76 | 0.99 | 1.16 | 1.29 | 1.35 | 0.40 | 0.52 | 0.61 | 0.67 | 0.71 |
| | V-Min | 1.73 | 2.10 | 2.34 | 2.49 | 2.61 | 0.81 | 0.98 | 1.10 | 1.18 | 1.22 | 0.42 | 0.51 | 0.58 | 0.62 | 0.65 |
| | 1F1B-R | 1.26 | 1.62 | 1.90 | 2.08 | 2.18 | 0.57 | 0.74 | 0.87 | 0.96 | 1.01 | 0.29 | 0.38 | 0.45 | 0.50 | 0.53 |
| MFU (%) | V-ZB | 50.4 | 55.7 | 57.9 | 59.1 | 61.3 | 50.5 | 56.1 | 58.4 | 59.8 | 61.2 | 45.6 | 53.9 | 56.0 | 57.1 | 58.5 |
| | ZB-1P | 43.5 | 50.2 | 55.2 | 58.2 | 60.0 | 43.5 | 50.3 | 55.2 | 58.9 | 60.5 | 40.9 | 47.8 | 52.6 | 55.7 | 57.7 |
| | V-Half | 38.3 | 47.3 | 53.1 | 56.8 | 59.8 | 38.1 | 47.4 | 53.4 | 57.0 | 59.5 | 36.2 | 45.2 | 51.1 | 54.6 | 57.0 |
| | 1F1B | 32.9 | 42.4 | 49.5 | 54.4 | 57.1 | 32.4 | 41.8 | 49.2 | 54.7 | 57.4 | 30.6 | 39.9 | 47.0 | 51.8 | 54.9 |
| | V-Min | 33.6 | 40.8 | 45.6 | 48.5 | 50.7 | 34.4 | 41.8 | 46.9 | 50.0 | 52.0 | 32.4 | 39.7 | 44.9 | 48.1 | 50.1 |
| | 1F1B-R | 24.5 | 31.5 | 36.9 | 40.5 | 42.4 | 24.2 | 31.3 | 36.8 | 40.8 | 42.9 | 22.8 | 29.7 | 35.0 | 38.6 | 40.9 |
| Peak memory (GB) | V-ZB | 58 | 59 | 59 | 59 | 59 | 57 | 57 | 57 | 57 | 57 | 55 | 55 | 55 | 55 | 55 |
| | ZB-1P | 58 | 57 | 57 | 57 | 57 | 57 | 56 | 56 | 56 | 56 | 55 | 55 | 55 | 55 | 55 |
| | V-Half | 38 | 38 | 38 | 38 | 38 | 38 | 38 | 38 | 38 | 38 | 38 | 38 | 38 | 38 | 38 |
| | 1F1B | 56 | 56 | 56 | 56 | 56 | 56 | 56 | 56 | 56 | 56 | 54 | 54 | 54 | 54 | 54 |
| | V-Min | 29 | 29 | 29 | 29 | 29 | 31 | 31 | 31 | 31 | 31 | 33 | 33 | 33 | 33 | 33 |
| | 1F1B-R | 14 | 14 | 14 | 14 | 14 | 18 | 18 | 18 | 18 | 18 | 24 | 24 | 24 | 24 | 24 |
| Activation memory (GB) | V-ZB | 48 | 48 | 48 | 48 | 48 | 43 | 43 | 43 | 43 | 43 | 36 | 36 | 36 | 36 | 36 |
| | ZB-1P | 48 | 47 | 47 | 47 | 47 | 43 | 42 | 42 | 42 | 42 | 36 | 36 | 36 | 36 | 36 |
| | V-Half | 28 | 28 | 28 | 28 | 28 | 24 | 24 | 24 | 24 | 24 | 19 | 19 | 19 | 19 | 19 |
| | 1F1B | 46 | 46 | 46 | 46 | 46 | 42 | 42 | 42 | 42 | 42 | 35 | 35 | 35 | 35 | 35 |
| | V-Min | 19 | 19 | 19 | 19 | 19 | 17 | 17 | 17 | 17 | 17 | 14 | 14 | 14 | 14 | 14 |
| | 1F1B-R | 1 | 1 | 1 | 1 | 1 | 1 | 1 | 1 | 1 | 1 | 1 | 1 | 1 | 1 | 1 |
| Bubble rate (%) | V-ZB | 18.7 | 8.88 | 4.57 | 2.32 | 1.16 | 18.9 | 8.51 | 4.39 | 2.19 | 1.09 | 18.7 | 8.61 | 4.3 | 2.24 | 1.14 |
| | ZB-1P | 31.7 | 20.1 | 11.2 | 5.92 | 3.06 | 31.5 | 19.8 | 11.0 | 5.82 | 3.0 | 32.1 | 20.1 | 11.1 | 5.89 | 3.04 |
| | V-Half | 40.5 | 24.2 | 13.8 | 7.41 | 3.84 | 41.3 | 24.6 | 14.0 | 7.5 | 3.89 | 42.0 | 24.9 | 14.3 | 7.6 | 3.97 |
| | 1F1B | 50.1 | 34.3 | 21.7 | 13.5 | 8.76 | 50.5 | 34.6 | 21.9 | 13.5 | 8.67 | 50.7 | 34.6 | 21.8 | 13.4 | 8.44 |
| | V-Min | 48.4 | 36.5 | 28.0 | 23.1 | 20.3 | 47.8 | 35.8 | 27.4 | 21.9 | 19.1 | 48.6 | 36.5 | 27.8 | 22.5 | 19.5 |
| | 1F1B-R | 63.0 | 51.2 | 41.9 | 35.8 | 32.2 | 63.3 | 51.3 | 41.9 | 35.7 | 32.0 | 63.5 | 51.6 | 42.0 | 35.8 | 32.2 |

## D.1    Comparing Pipeline Schedules

For Section 4.2, we present our detailed experiment data in Table 4. Specifically, the metrics are defined as:

- MFU: The FLOPS utilization of the training system. The calculation of FLOPS of a model is following [Narayanan et al., 2021].
- Peak Memory: The maximum peak memory cross all devices.

- Activation Memory: Estimated as deducting the iteration-start memory from peak memory on each device. The number presented is the maximum activation memory cross all devices.
- Bubble Rate: The theoretical bubble rate reported by scheduler, using profiled run times of $F$, $B$ and $W$ at starting iterations.

## D.2 Single-pass MFU Gain When Increasing Microbatch Size

To evaluate how increasing microbatch size increases the arithmetic intensity, we profile the run time of each single F/B/W pass and calculate their single-pass MFU. We list the results in Table 5. It shows that whether there are significant MFU gain depends on both the model and the microbatch size.

Table 5: Single-pass MFU gain when increasing microbatch size

| Model | 9.6B | | | 21B | | | 38.5B | | |
|---|---|---|---|---|---|---|---|---|---|
| Microbatch Size | 4 | 8 | 12 | 2 | 4 | 6 | 1 | 2 | 3 |
| F Pass (ms) | 12.96 | 26.30 | 39.45 | 9.30 | 18.11 | 26.81 | 6.72 | 12.27 | 18.05 |
| B Pass (ms) | 13.22 | 26.66 | 39.85 | 9.47 | 18.55 | 27.09 | 6.89 | 12.84 | 18.73 |
| W Pass (ms) | 9.76 | 19.62 | 28.93 | 7.19 | 14.03 | 21.82 | 5.06 | 9.63 | 15.46 |
| FBW Average MFU (%) | 72.13 | 71.45 | 71.86 | 71.11 | 72.83 | 73.14 | 66.86 | 71.87 | 71.69 |

## D.3 More Details on Grid Search

Table 6: MFU of grid search, with $SequenceLength = 1024$ and $BatchSize = 160$

| Parallelization | MicroBS | 1F1B | 1F1B-R | ZB-1P | V-Half | V-Min | V-ZB |
|---|---|---|---|---|---|---|---|
| DP=1 | 1 | 51.66 | 38.18 | 52.59 | 52.32 | 47.36 | **53.42** |
| TP=4 | 2 | 54.0 | 40.32 | 56.37 | 55.92 | 50.24 | **58.25** |
| PP=10 | 4 | 52.44 | 38.81 | - | **55.49** | 49.79 | - |
| DP=2 | 1 | 54.17 | 40.84 | 57.26 | 56.85 | 52.8 | **59.03** |
| TP=2 | 2 | - | 40.35 | - | **57.03** | 52.3 | - |
| PP=10 | 4 | - | **35.14** | - | - | - | - |
| DP=1 | 1 | 54.03 | 40.55 | 57.65 | 56.88 | 50.61 | **59.59** |
| TP=2 | 2 | 53.3 | 39.83 | 57.55 | 56.9 | 50.4 | **60.23** |
| PP=20 | 4 | - | 34.62 | - | - | **45.52** | - |
| DP=2 | 1 | - | 40.34 | - | **57.83** | 52.78 | - |
| TP=1 | 2 | - | 35.89 | - | - | **48.83** | - |
| PP=20 | 4 | - | **27.78** | - | - | - | - |
| DP=1 | 1 | 53.47 | 40.07 | 59.95 | 57.68 | 52.54 | **63.31** |
| TP=1 | 2 | - | 35.69 | - | **54.08** | 48.27 | - |
| PP=40 | 4 | - | **27.53** | - | - | - | - |
| DP=1 | 1 | 44.71 | 33.21 | 44.6 | 44.05 | 36.88 | **45.32** |
| TP=8 | 2 | 51.75 | 38.67 | 52.67 | 51.94 | 45.37 | **53.27** |
| PP=5 | 4 | 52.0 | 38.49 | 53.84 | 52.98 | 46.67 | **53.98** |
| | 8 | 50.2 | 36.84 | - | **51.87** | 46.83 | - |
| DP=2 | 1 | 51.56 | 38.73 | 53.38 | 52.54 | 48.7 | **53.6** |
| TP=4 | 2 | 54.77 | 40.69 | 57.48 | 56.32 | 52.24 | **58.05** |
| PP=5 | 4 | - | 39.61 | - | **56.53** | 52.16 | - |

We show the MFU of every setup of our grid search in Table 6, 7 and 8 for three groups of experiments: one with $SequenceLength = 1024$ and $BatchSize = 160$, one with $SequenceLength = 3072$ and $BatchSize = 640$ and the other with $SequenceLength = 16384$ and $BatchSize = 160$.

For the first experiment group, the best setup is *V-ZB* under pure PP because of its bubble elimination. For the second setup, the best setup is *V-Half* because its memory efficiency enables a lower TP

degree, which is otherwise impossible for *V-ZB*/ZB-1P/1F1B. For the last setup, due to high memory pressure only *V-Min* and *V-Half* can run without checkpointing. A comparison of TP and PP can be found at D.4.

Table 7: MFU of grid search, with $SequenceLength = 3072$ and $BatchSize = 640$

| Parallelization | MicroBS | 1F1B | 1F1B-R | ZB-1P | V-Half | V-Min | V-ZB |
|---|---|---|---|---|---|---|---|
| DP=1 | 1 | 62.06 | 45.85 | 62.18 | 62.17 | 56.29 | **62.56** |
| TP=4 | 2 | - | 45.52 | - | 45.86 | **55.54** | - |
| PP=10 | 4 | - | **45.34** | - | - | - | - |
| DP=2 | 1 | - | 49.03 | - | - | **60.59** | - |
| TP=2 | 2 | - | **47.39** | - | - | - | - |
| PP=10 | 4 | - | **45.35** | - | - | - | - |
| DP=1 | 1 | - | 48.63 | - | **66.34** | 57.92 | - |
| TP=2 | 2 | - | **47.18** | - | - | - | - |
| PP=20 | 4 | - | **45.3** | - | - | - | - |
| DP=2 | 1 | - | **50.37** | - | - | - | - |
| TP=1 | 2 | - | **46.01** | - | - | - | - |
| PP=20 | 4 | - | - | - | - | - | - |
| DP=1 | 1 | - | 49.7 | - | - | **61.04** | - |
| TP=1 | 2 | - | **45.26** | - | - | - | - |
| PP=40 | 4 | - | **42.73** | - | - | - | - |
| DP=1 | 1 | **55.93** | 41.31 | 54.13 | 54.22 | 49.75 | 54.38 |
| TP=8 | 2 | **57.64** | 42.37 | 55.25 | 55.64 | 51.88 | 55.76 |
| PP=5 | 4 | - | **43.3** | - | - | 14.59 | - |
| DP=2 | 1 | **62.95** | 46.49 | - | 61.95 | 57.74 | - |
| TP=4 | 2 | - | **45.92** | - | - | - | - |
| PP=5 | 4 | - | **44.44** | - | - | - | - |

Table 8: MFU of grid search, with $SequenceLength = 16384$ and $BatchSize = 160$

| Parallelization | MicroBS | 1F1B | 1F1B-R | ZB-1P | V-Half | V-Min | V-ZB |
|---|---|---|---|---|---|---|---|
| DP=1;TP=4;PP=10 | 1 | - | **42.05** | - | - | - | - |
| DP=2;TP=2;PP=10 | 1 | - | - | - | - | - | - |
| DP=1;TP=2;PP=20 | 1 | - | **41.52** | - | - | - | - |
| DP=2;TP=1;PP=20 | 1 | - | - | - | - | - | - |
| DP=1;TP=1;PP=40 | 1 | - | - | - | - | - | - |
| DP=1;TP=8;PP=5 | 1 | - | 39.48 | - | **57.85** | 48.58 | - |
| DP=1;TP=8;PP=5 | 2 | - | - | - | - | - | - |
| DP=2;TP=4;PP=5 | 1 | - | **35.13** | - | - | - | - |

## D.4 Model Parallelism: More PP or More TP?

Our grid search results in Appendix D.3 show a strong favor of Pipeline Parallel (PP) over Tensor Parallel (TP), which might contradict with some existing industry experience where more degree of TP usually accelerates training. To understand the reason, we briefly compare TP and PP in this section.

Though PP also equally partition the model into $p$ PP shards, it usually needs to cache the activations for $\Theta(p)$ microbatches, resulting in the total activation memory demand same as the unpartitioned. On the other hand, TP, when used with sequence parallelism [Korthikanti et al., 2023], partitions most activation memory equally to $t$ TP shards, which is one of the most significant benefit of TP over PP. However, this comes at the cost of a collective communication and reducing the size of hidden dimension to $\frac{1}{t}$, which can significantly decrease the single-pass (F/B/W) MFU. Though one can argue that the saved memory can be used to increase the microbatch size, our experiment measuring the MFU under different TP setups (Figure 12) demonstrates that a higher-degree of TP even with larger microbatch size still suffers from lower single-pass MFU.

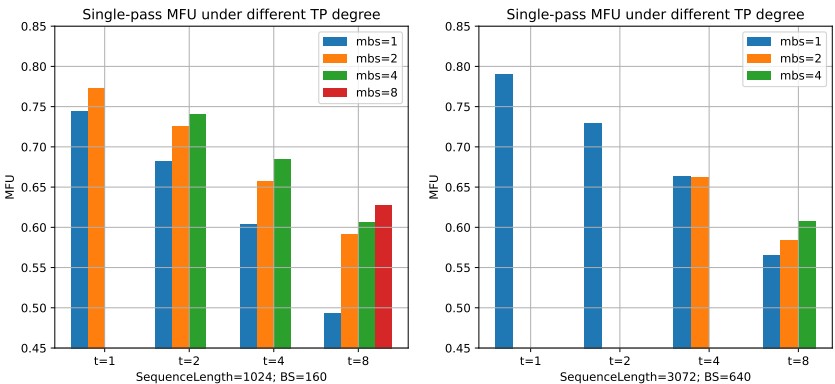

Figure 12: Average single-pass MFU (over FBW) for grid search under different TP degrees.

The throughput of PP also decreases as PP degree increases for two major reasons: a) for PP schedules with a fixed global batch size, a higher PP degree usually results in higher pipeline bubbles. For example, 1F1B has a bubble rate of $\frac{p-1}{p+n-1}$, which would increase if $p$ grows and $n$ keeps unchanged; b) the first and last pipeline stage for a language model usually have an embedding or output layer which has innegligible additional computations, making the pipeline unbalanced and hurting the efficiency. With higher PP degree, the unbalance would be aggravated. However, theses two issues can be mitigated. For the first issue, a larger number of microbatches or using *V-ZB* can significantly reduce the pipeline bubble. For the second issue, we can use another standalone stage to handle the embedding or output layer instead to avoid bottleneck. In our experiments, we simply deduct 1 layer from both the first and last pipeline stage to mitigate this problem. As a result, our methods essentially push the preference from TP to PP.

## E  Bubble Analysis

Although we mainly focus on the memory efficiency in this paper, we also need to take care of the bubbles in pipeline parallelism. Typically, there is a trade-off between memory and bubble, where allowing more memory usage can reduce the bubble of a pipeline schedule. In this section, we will discuss how to identify whether the bubble of a schedule will grow with respect to the number of microbatches. Additionally, we illustrate an empirical lower bound of bubble in our methods. We follow the notation in Section 3.1 and Appendix A in this section.

### E.1  Existence of Repeating Bubbles

In pipeline parallelism, we usually expect there is no repeating bubbles introduced in Section 3.3, namely, the bubble is only related to the number of pipeline devices ($d$), and won't grow when increasing the number of microbatches ($n$). This property guarantees a good scalability with respect to $n$. We say a pipeline schedule is with $O(d)$ bubble if it satisfies this property, otherwise with $O(n)$ bubble. Note that the conclusion is based on the values of $T_F, T_B$ and $T_W$, which are the run times of $F$, $B$ and $W$ respectively. For example, the schedule of *V-Min* (Figure 4b) is with $O(d)$ bubble when $T_F = T_B = T_W$, but is with $O(n)$ bubble when $T_W$ is significantly smaller than $T_F$ and $T_B$.

**If there are no repeating bubbles in a schedule for any values of $T_F, T_B, T_W$, the minimal memory is $2dm$.**  In a pipeline schedule, there are two types of dependencies, streaming dependency within the same device and computational dependency across devices. We define a dependency path as a sequence of passes $\tau = (v_1, v_2, ..., v_{|\tau|})$ where for any $1 < i \leq |\tau|$, $v_i$ is dependent on $v_{i-1}$ (either streaming dependency or computational dependency). We define the time cost of a dependency path as $T_\tau = \sum_{1 \leq i \leq |\tau|} T_{v_i}$ where $T_{v_i}$ is the time cost of $v_i$. Then the runtime of the whole pipeline should be $T = \max_\tau T_\tau$.

Obviously, there is a trivial dependency path $\tau_1$ that all the passes are from the same device, and $T_{\tau_1} = 2n(T_F + T_B + T_W)$. Note that it is the lower bound of runtime, and any extra runtime would be considered as bubbles.

Let's consider another dependency path $\tau_2$ containing only forward passes. To be simple, we denote $\hat{F}_j$ as the forward sequence for the $j$-th microbatch. Then $\tau_2 = concatenate(\hat{F}_0, \hat{F}_k, ..., \hat{F}_{\lfloor \frac{n-1}{k} \rfloor k})$ thus $T_{\tau_2} = 2d\lfloor \frac{n+k-1}{k} \rfloor T_F$, where $6k > \delta(F_1^0, F_1^1) > 6(k-1)$ (greedily include as many forward passes as possible). Note that if we want to guarantee $O(d)$ bubble for any values of $T_F, T_B, T_W$, we should choose $k \geq d$ to make $2d\lfloor \frac{n+k-1}{k} \rfloor \leq 2n$, otherwise $2d\lfloor \frac{n+k-1}{k} \rfloor T_F - 2n(T_F + T_B + T_W) \in O(n)$ if we set $T_B = T_W = 0$. Then we can get $\delta(F_1^0, F_1^1) > 6d - 6$.

Then we consider a similar dependency path $\tau_3$ containing only backward passes, and we can get $\delta(B_1^1, B_1^0) > 6d - 6$. According to Equation 1, we can get the peak memory in the first device is at least $(\lceil \frac{\delta(F_1^0, F_1^1)}{6} \rceil + \lceil \frac{\delta(B_1^1, B_1^0)}{6} \rceil)m \geq 2dm$.

**For most real cases, *V-Half* is enough to guarantee there are no repeating bubbles.** Although the above proof shows that $2dm$ memory is required to guarantee $O(d)$ bubble for any values of $T_F, T_B, T_W$, we don't need that much memory in practice because the values of $T_F, T_B$ and $T_W$ are well constrained. As in Qi et al. [2023], $F$, $B$ and $W$ have similar total FLOPS, and $T_F, T_B, T_W$ don't differ too much in real cases. Based on this insight, we can check the conditions where our methods are with $O(d)$ bubble.

Because both warm-up phase and cool-down phase have $O(d)$ passes, we only need to consider the stable phase to identify whether a schedule is with $O(d)$ bubble or with $O(n)$ bubble. Obviously, streaming dependency won't block the device from executing the next pass. So bubble is caused only by the computational dependency. Formally, a schedule is with $O(n)$ bubble if and only if there exist two passes $u$ and $v$ (within the same device) and there are two dependency paths $\tau$ and $\tau'$ between them, where $\tau$ only contains streaming dependencies, $\tau'$ contains at least two computational dependencies, and $T_\tau < T_{\tau'}$. Based on our V-shape building blocks, we only need to check $u$ and $v$ with a small distance ($< 6$) and $\tau'$ within two adjacent devices. In this way, we can conclude that the schedule in Figure 4c is with $O(d)$ bubble when $T_W + 2T_B \geq 2T_F$ & $T_W + 2T_F \geq 2T_B$, which is satisfied in most real cases.

### E.2 Lower Bound

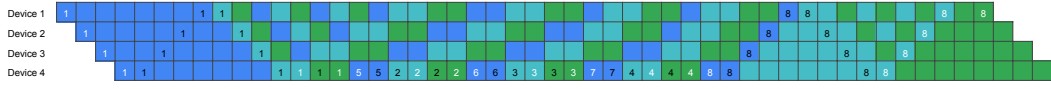

Figure 13: A dependency path of pipeline schedule.

The runtime of a schedule can be bounded by any dependency path. In Figure 13, we present a dependency path which is a non-trivial lower bound for most schedules after squeezing and reordering. Assuming the actual peak memory is $km$ ($k \leq 2d$) and $T_F = T_B = T_W = 1$, then the runtime of this dependency path is at least $6n + 6d - 3k - 1$. Empirically, we find that we can always get close to this lower bound in our adaptive scheduler.

## F  Building blocks of *V-Min* and *V-Half* for all values of $d$

To avoid collisions when repeating building blocks, we need a slightly different offsets for different number of devices $d$. We list the details in Table 9 and 10 while continue using the notation defined in A. We do not list the offsets related to $W$ passes because they always fill in the empty grids left by $F$ and $B$. Some samples can also be found at Figure 14.

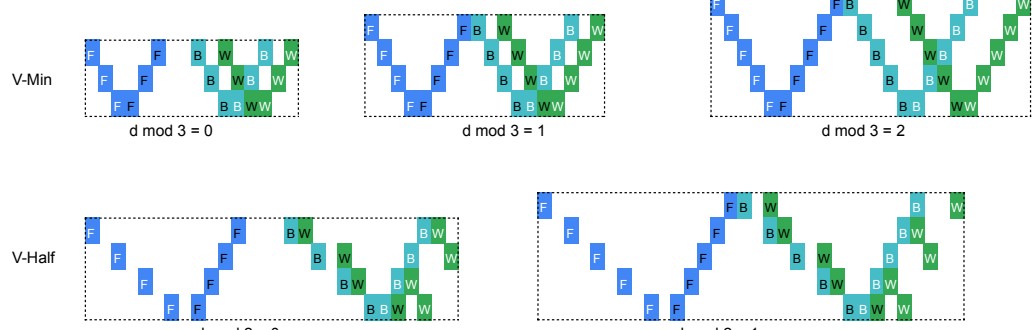

Figure 14: Building blocks of *V-Min* and *V-Half* on different settings of $d$

Table 9: Offsets for *V-Min*

| $1 \le i < d, 1 < j \le d$ | $\delta F_i^0$ | $\delta F_j^1$ | $\delta B_i^0$ | $\delta B_j^1$ | $\delta(F_d^0, F_d^1)$ | $\delta(F_1^1, B_1^1)$ | $\delta(B_d^0, B_d^1)$ |
|---|---|---|---|---|---|---|---|
| $d \equiv 0 \mod 3$ | 1 | 1 | 1 | 1 | 1 | 3 | 1 |
| $d \not\equiv 0 \mod 3$ | 1 | 1 | 1 | 1 | 1 | 1 | 1 |

Table 10: Offsets for *V-Half*

| $1 \le i < d, 1 < j \le d$ | $\delta F_i^0$ | $\delta F_j^1$ | $\delta B_i^0$ | $\delta B_j^1$ | $\delta(F_d^0, F_d^1)$ | $\delta(F_1^1, B_1^1)$ | $\delta(B_d^0, B_d^1)$ |
|---|---|---|---|---|---|---|---|
| $d \equiv 0 \mod 2$ | 2 | 1 | 2 | 1 | 2 | 4 | 1 |
| $d \equiv 1 \mod 2$ | 2 | 1 | 2 | 1 | 2 | 1 | 1 |

# G Non-uniform Repeating Interval of Interleaved 1F1B

While most existing schedules (Figure 18) are repeated with uniform interval, interleaved 1F1B [Narayanan et al., 2021] is slightly different in the repeating pattern. The official interleaved 1F1B has a repeating pattern as shown in Figure 15a. If we also employ uniform repeating interval as in Figure 15b, we can obtain another schedule with the same memory footprint and bubble rate as the official interleaved 1F1B, shown in the gallery (Figure 18i).

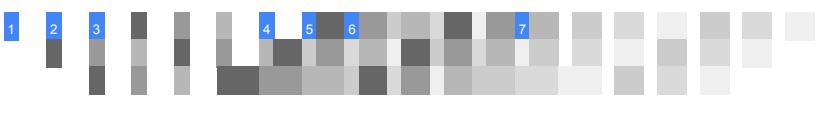

(a) The non-uniform repeating interval of interleaved 1F1B. The number in highlighted grids shows the microbatch with that index starts at this cell.

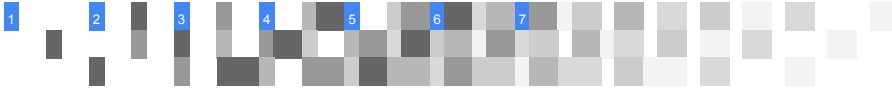

(b) A variation of interleaved 1F1B with uniform repeat.

Figure 15: Different repeat pattern of the same building block of interleaved 1F1B.

# H Other Memory Efficient Building Blocks

Under the guidance of lifespan defined in Section 2.2, we also find some other building blocks besides the V-Shape building block family. We show the building blocks in Figure 16 and their full schedules

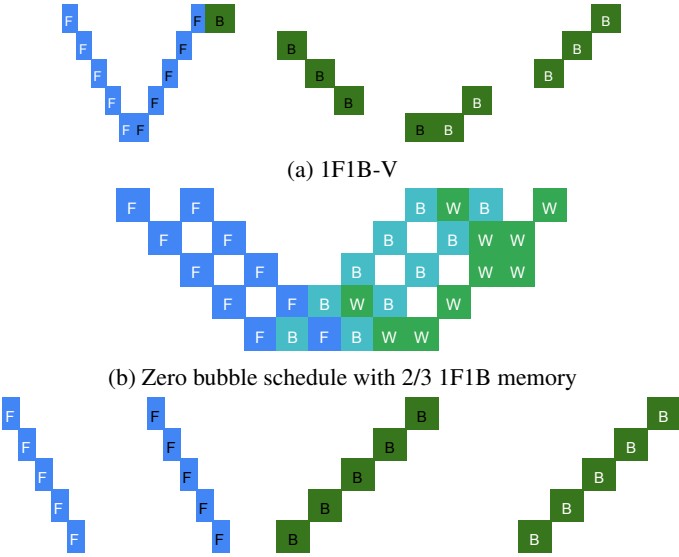

(a) 1F1B-V

(b) Zero bubble schedule with 2/3 1F1B memory

(c) A variation of interleaved 1F1B with the same bubble rate but lower memory

Figure 16: Other memory-efficient building blocks.

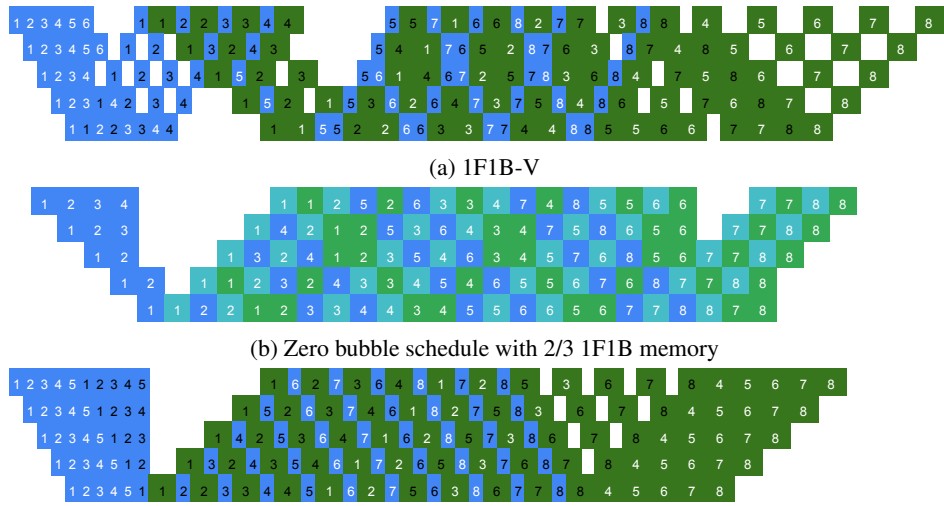

(a) 1F1B-V

(b) Zero bubble schedule with 2/3 1F1B memory

(c) A variation of interleaved 1F1B with the same bubble rate but lower memory

Figure 17: Other memory-efficient schedules.

in Figure 17. All these schedules have lower bubble rate than 1F1B. Specifically, 1F1B-V applies V-Shape to the building block of 1F1B but without *B-W* split, which can reduce the peak memory to asymptotically 2/3 of 1F1B. We also find that utilizing *B-W* split, the zero bubble pipeline schedules proposed in [Qi et al., 2023] with configurable memory limit can support a minimum of 2/3 activation memory of 1F1B, using the building block shown in Figure 16b. Note that two microbatches are included in a single building block to avoid collision. Using the building block defined in Figure 16c, we can make a schedule with the same bubble rate as interleaved 1F1B but lower memory, shown in Figure 17c.

# I   A Gallery of Pipeline Parallel Schedules and Their Building Blocks

We show the building blocks and full schedules of some well-known existing methods in Figure 18.

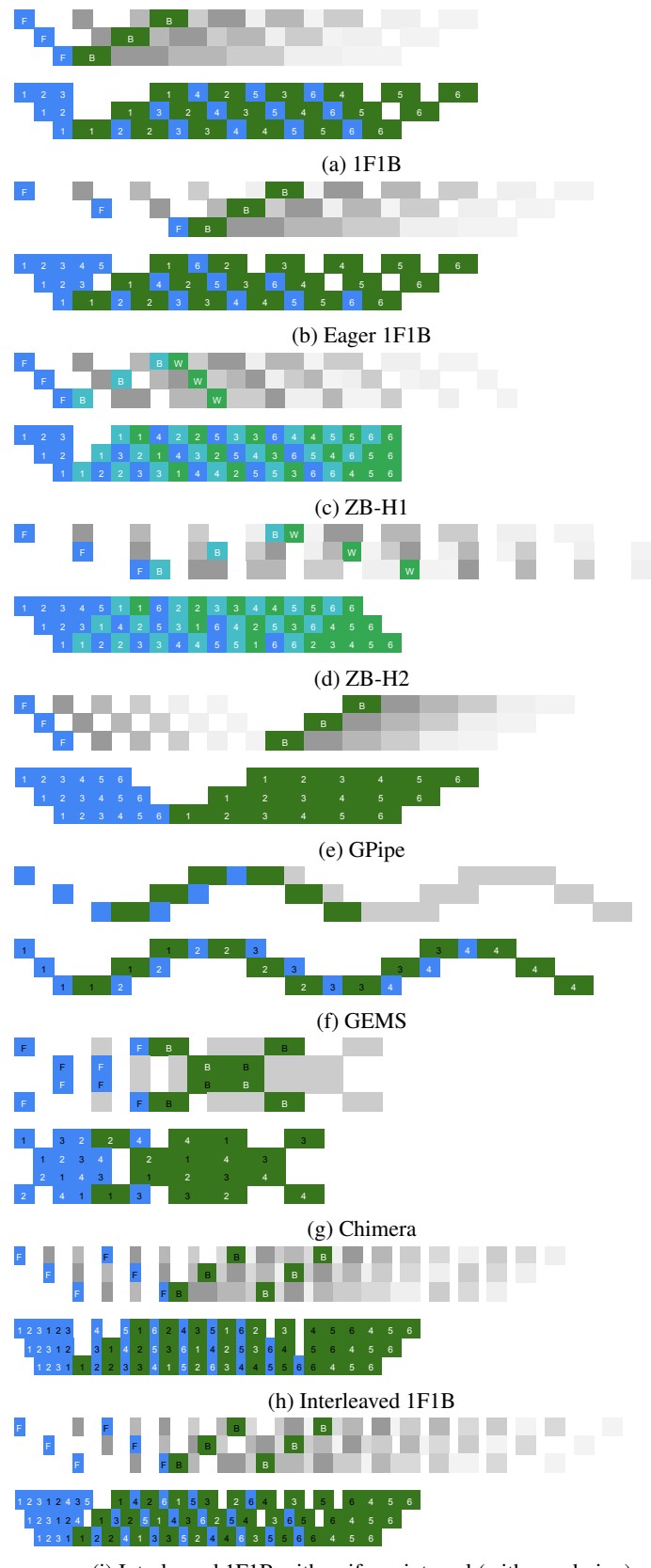

(a) 1F1B

(b) Eager 1F1B

(c) ZB-H1

(d) ZB-H2

(e) GPipe

(f) GEMS

(g) Chimera

(h) Interleaved 1F1B

(i) Interleaved 1F1B with uniform interval (with reordering)

Figure 18: A gallery of pipeline schedules and their building blocks. The upper row of each schedule shows the building block and how it repeats. The lower row shows the final schedule after squeezing.

