# OpenReview forum: "Pipeline Parallelism with Controllable Memory"
_NeurIPS.cc/2024/Conference — NeurIPS 2024 poster_

### Official Review · Reviewer_b8L7 · 2024-07-07

**Soundness:** 3
**Presentation:** 3
**Contribution:** 3
**Rating:** 7
**Confidence:** 4

**Summary:**

The paper addresses the inefficiencies in current pipeline parallelism schedules used for training large-scale models, particularly focusing on high activation memory and pipeline bubbles. The paper proposes a new framework to decompose pipeline schedules into repeating building blocks and introduces V-shape building blocks (V-Min, V-Half, V-ZB) with controllable activation memory. Such building blocks reduce peak activation memory compared to the traditional 1F1B approach and even achieve higher throughput with reduced pipeline bubbles. The evaluations demonstrate substantial improvements in throughput and memory efficiency over 1F1B and 1F1B-I methods.

**Strengths:**

- The authors decompose pipeline parallelism into repeating building blocks, providing a systematic methodology for designing more efficient schedules. This approach helps in understanding and optimizing pipeline parallelism in a structured manner.
- The V-Shape building blocks demonstrate the structured understanding of pipeline parallelism.
- The authors provide experimental evaluations demonstrating the practical benefits of the proposed methods.
- By reducing pipeline bubbles, the paper demonstrates that PP can become a much more preferred option over TP at practical large-scale training.

**Weaknesses:**

- It would be helpful to have experimental results for long sequence lengths. 1024 and 3072 sequence lengths are too short compared to what SOTA LLMs can handle.

**Questions:**

- One of the problems with pipeline parallelism is that it increases implementation complexity because of its multi-program-multi-data (MPMD) nature. Do V-shaped abstractions bring any convenience in terms of implementation perspective?

**Limitations:**

No societal impact exists.

---

> ### Author Rebuttal · Authors · 2024-08-06
>
> We thank the reviewer for the positive feedback and suggestions for improvement. We respond to individual points from your review below.
>
> >It would be helpful to have experimental results for long sequence lengths. 1024 and 3072 sequence lengths are too short compared to what SOTA LLMs can handle.
>
> Thanks for pointing out this. Theoretically, increasing the sequence lengths should not significantly affect our conclusions, as our methods are flexible with different F/B/W timings. On the contrary, increasing the sequence length will increase memory pressure, highlighting the advantages of our memory-saving schedules.
>
> We added another set of grid-search experiment using almost the same setup but with sequence length=16384. Here is a summary of experiment result. In this setup, 1F1B, ZBV and ZB1P run OOM for all combination of distributed training parameters (dp/tp/pp degrees). The experiment shows V-Half is 38% faster than 1F1B-R, which is the only baseline method that can run.
>
> | Common Setup | PP Method | Best MFU (%) | Best Parameters |
> | ------ | ------ |------ | ------ |
> |  | 1F1B | OOM | - |
> | 98.5B Model | 1F1B -R | 42.05% | dp=1;tp=4;pp=10;microbs=1 |
> | Seq Length 16384 | ZB1P | OOM | - |
> | Batch Size 160 | V-Half | 57.85% | dp=1;tp=8;pp=5;microbs=1 |
> | | V-Min | 48.58% | dp=1;tp=8;pp=5;microbs=1 |
> | | V-ZB | OOM | - |
>
>
> >One of the problems with pipeline parallelism is that it increases implementation complexity because of its multi-program-multi-data (MPMD) nature. Do V-shaped abstractions bring any convenience in terms of implementation perspective?
>
> Thanks for discussing this question from the implementation perspective. We believe that implementing pipeline parallelism primarily involves two components: a) a scheduler that determines the order of passes, and b) a runtime that executes the schedule.
>
> Regarding the scheduler, we believe the V-Shaped abstractions simplify the implementation. Developers only need to implement the building blocks, and we can use a framework to handle automated repeating, squeezing, and reordering. This approach is exactly how we implemented the schedules in our experiments.
>
> For the runtime, implementing a pipeline parallelism (PP) runtime is indeed more complex. It requires careful insertion of communications between stages to avoid performance degradation (e.g., stalled send/recv operations) or deadlocks. Note that this complication is inherent to PP and not introduced by V-shape schedules. To address this, we built a uniform runtime that automatically inserts and fuses (or refrains from fusing when better) communication operations according to the schedule produced by the scheduler. This allows the scheduler to focus solely on the ordering of computation passes. We will also open-source our PP runtime in hopes that it will benefit the community.

---

### Official Review · Reviewer_w4r8 · 2024-07-10

**Soundness:** 3
**Presentation:** 2
**Contribution:** 3
**Rating:** 7
**Confidence:** 3

**Summary:**

This work proposes systematic methodology for designing pipeline parallelism schedules and analyzing their performance (e.g. peak memory and pipeline bubbles).
The major observation is that a pipeline schedule can be viewed as repeating a building block, and the peak memory critically depends on the lifespan of a block.
Based on these insights, the authors design a family of novel V-shape building blocks and pipeline schedules that are memory efficient and achieve higher throughput than strong baselines like 1F1B (though at the cost of more communication), which is validated both theoretically and empirically.

**Strengths:**

- This work proposes a systematic and unified perspective for pipeline parallelism schedule,
with some novel insights that might foster the design of better schedules in the future.
- Analysis and experiments are extensive and well executed, confirming the efficacy of the proposed methodology.
- Overall, writing and presentation are clear.

**Weaknesses:**

There is still room for improvement in writing. For example:
- There are quite some typos, some of which are listed here: Line 22 "tensor" -> "Tensor"; Line 189, "V-Min" -> "V-ZB" (?); Line 193, "serious" -> "series".
- Line 210, "while other methods' memory is similar": one exception is 1F1B-R, whose activation memory is much smaller.

**Questions:**

A high-level question that might be worth a brief discussion in the manuscript is:
is it true that a pipeline schedule *must be* repetition of a building block?
Actually the answer is no.
A naive example I can think of is a variant of GPipe, where the order of backward passes can be arbitrary, and thus the building block for one microbatch can be different from that of another microbatch.
This extension of GPipe is designed just out of theoretical curiosity though.

So the next question is whether there could be any real advantage (in terms of peak memory, throughput, etc.) in a pipeline schedule that cannot be expressed as repetition of a building block.
It would be great if there is a clear answer with a formal proof.
But even if the authors do not know the answer yet (and neither do I), it might be worth mentioning, so that anyone who has read this work will keep this in mind, rather than blindly constrain themselves to repeating a building block when designing new schedules.

**Limitations:**

Limitations are discussed throughout the manuscript.

---

> ### Author Rebuttal · Authors · 2024-08-06
>
> We thank the reviewer for the valuable feedback and suggestions for improvement. We respond to individual points from your review below.
>
> >There are quite some typos, some of which are listed here: Line 22 "tensor" -> "Tensor"; Line 189, "V-Min" -> "V-ZB" (?); Line 193, "serious" -> "series".
> Line 210, "while other methods' memory is similar": one exception is 1F1B-R, whose activation memory is much smaller.
>
> Thanks for the careful reading. We have corrected them in the revised version. We also used a grammar tool to check the whole paper and correct the grammar errors.
>
> >A high-level question that might be worth a brief discussion in the manuscript is: is it true that a pipeline schedule must be repetition of a building block? Actually the answer is no. A naive example I can think of is a variant of GPipe, where the order of backward passes can be arbitrary, and thus the building block for one microbatch can be different from that of another microbatch. This extension of GPipe is designed just out of theoretical curiosity though.
> So the next question is whether there could be any real advantage (in terms of peak memory, throughput, etc.) in a pipeline schedule that cannot be expressed as repetition of a building block. It would be great if there is a clear answer with a formal proof. But even if the authors do not know the answer yet (and neither do I), it might be worth mentioning, so that anyone who has read this work will keep this in mind, rather than blindly constrain themselves to repeating a building block when designing new schedules.
>
> This is very true, thank you for bringing up this insightful point. We add a sentence "Notice that repeating a building block is not the only way of building a pipeline, other methods like greedy search could generate a pipeline that has no repeating patterns." in Conclusion.
>
> Some more discussions on the points that may favor repeating building blocks.
> 1. Repeating the building blocks means uniform lifespans, and intuitively it is better than unbalanced lifespans where a single point could cause a larger peak memory.
> 2. Mentally more tractable, the complexity of designing a "good" pipeline is reduced to designing a "good" building block.
> 3. Additionally, the scalability with respect to the number of microbatches is guaranteed.

---

> > ### Comment · Reviewer_w4r8 · 2024-08-12
> >
> > Thank you for the rebuttal. I have no further question, and will keep my score.

---

### Official Review · Reviewer_vShK · 2024-07-13

**Soundness:** 3
**Presentation:** 2
**Contribution:** 3
**Rating:** 5
**Confidence:** 3

**Summary:**

Authors propose a way to identify the repeated building block a pipeline schedule is built from. By relating the peak activation memory of the schedule to the lifespan of the building block the authors show that existing schedules do not optimally use memory, and design higher-throughput schedules that use the same activation memory as the baseline. The methods demonstrate up to 55% improvement in throughput for pure pipeline parallelism settings and 16% improvement for large language models over the 1F1B baseline. The paper also provides a systematic methodology for designing pipeline schedules, emphasizing memory usage balance and reduced pipeline bubbles.

**Strengths:**

The paper introduces a novel framework for decomposing pipeline parallelism schedules into memory-efficient building blocks. This approach addresses inefficiencies observed in existing schedules. The systematic methodology for designing pipeline schedules emphasizes memory usage balance and reduced pipeline bubbles.

Discussion of peak memory and new memory efficient building blocks is the core of the paper and the part which is most clear. The authors provide technical details, including the proposed building blocks, their implementation, and experimental results.

**Weaknesses:**

The paper suffers from quite a few clarity issues and could use some copyediting (there are lots of small grammatical errors, etc.). Main clarity issues in the paper are, for example, a lack of useful captions in figures, a lack of substantive discussion in some of the appendices (i.e., “we leave the discussion to appendix X” but there is not much discussion in appendix X).

A lot could be done to improve clarity of the discussion in section 3. For example, although the paper explicitly details asymptotic behavior (d -> infinity?) this is really only mentioned in lines 166-167, which somewhat confuses the issue. Some brief discussion of what the effect is in low-d situations would be nice (not new experiments – just a qualitative idea). Fig. 3 and Table 1 contradict each other due to the presence/absence of the asymptotic limit, which is mentioned in the title of Table 1 and briefly in the text but is not very clear, especially since Fig. 3 and Table 1 are supposed to be read together (?).

The figure captions should at minimum restate what is shown in the figures and what relevant terms (d, l, etc.) mean – this makes it much easier to refer to the figure without searching through the text for definitions and explanations. This should be doable in 1 or 2 sentences per caption at most and should not take a lot of space.

Some plots are hard to read. For example, Figure 4 and Figure 5 shows detailed pipelines using various colors of blocks and fonts without proper definition or explanation. It is also not clear the shown pipeline is the actual setting or a high-level demonstration of the design.

V-Half seems to be a heuristic method based on V-Min and V-ZB. It may not be an optimal solution for the pipeline. Other configurations regarding the trade-off of memory and throughput are not evaluated.

Section 4.4 is confusing. Table 3 is not referenced anywhere and not explained in this section.  The results mentioned in this section (Table 6) do not clearly show the combined performance of the proposed approach and existing techniques.

**Questions:**

Why is 1F1B the baseline used for the paper? It feels like there could be some more discussion here that is missing.

Is the lack of bubble in V-Half (figure 5) purely empirical or is there some reason it should be expected? There doesn’t seem to be anything special about V-Half from the point of view of the discussion in the paper (only the activation memory is relevant to V-half; it’s not clear why this specifically should not have bubbles). Also, V-ZB has no bubbles in the case of differing runtimes, if I understand correctly; so maybe it is more accurate to say that the number of bubbles decreases to 0 as activation memory budget increases?

How is the number of stages in the pipeline determined? While the authors mentioned that it depends on the number of devices, what is the number of stages used in this work? How does the number of stages influence performance?

Given the constraints of memory or throughput, do the authors use grid search to find an optimal schedule? Is it possible to use a theoretical or approximation method to quickly find a good pipeline schedule?

**Limitations:**

Yes

---

> ### Author Rebuttal · Authors · 2024-08-06
>
> We thank the reviewer for the valuable feedback and insightful suggestions for improvement. We have revised the manuscript to address most of the mentioned issues. We respond to individual points from your review below.
>
> >grammatical errors, a lack of useful captions in figures, Some plots are hard to read.
>
> * We have used the grammar tool to check and correct the grammar errors.
> * We add captions for most of the figures to summarize the key points or conclusions, explain the notations in Figure 3 and Table 1.
> * We re-organize the layout of figures to make it more friendly to readers.
>
> >a lack of substantive discussion in some of the appendices.
>
> Could you provide specific pointers?
>
> >A lot could be done to improve clarity of the discussion in section 3... the effect in low-d situations...
>
> To make it more clear, we rewrote the Section 3.1 to include more technical details in the main paper (refer to global response). Specifically,
> * We use uniform offsets between adjacent F/B passes across different devices, and leave those within the same devices flexible. This is where the asymptotic behavior comes from.
> * The exact peak memory is $\lceil \frac{d+2}{3} \rceil \frac{M}{d}$ for V-Min, and $\lceil \frac{d+1}{2} \rceil \frac{M}{d}$ for V-Half. Actually we intended to mention it in the paper, somehow we forgot to do so, thanks for pointing this out.
>
> >not clear the shown pipeline is the actual setting or a high-level demonstration of the design.
>
> The pipeline shown in Figure 4 is the actual setting with 4 devices and 8 microbatches. (Refer to the PDF in global response)
>
> >V-Half seems to be a heuristic method... It may not be an optimal... Other configurations... are not evaluated.
>
> V-Half is not heuristically designed, all schedules are searched systematically with designated memory limit. There are indeed some other configurations from the search, e.g. a pipeline with 2/3 memory and 1/3 bubbles of 1F1B. As V-Min and V-ZB are optimal in memory and throughput, V-Half is to represent something in the middle. For the trade-off of memory and throughput, please refer to Figure 9 and 10 in the Appendix.
>
> >Section 4.4 is confusing. Table 3 is not referenced anywhere... (Table 6) do not clearly show...
>
> It's a mistake due to duplicated label name in latex, it is fixed now, thanks for spotting it.
>
> >Why is 1F1B the baseline used for the paper? It feels like there could be some more discussion here that is missing.
>
> Instead of saying using 1F1B as a baseline, we are more of using it as a reference point. The memory consumption of other methods are normalized to a proportion of 1F1B, simply because 1F1B is the most well known schedule that readers should be familiar with.
>
> >Is the lack of bubble in V-Half purely empirical or ... expected? ... Also, V-ZB has no bubbles ... differing runtimes, ... so maybe it is more accurate to say that the number of bubbles decreases to 0 as activation memory budget increases?
>
> V-Half is free of **repeating bubbles** (in most cases), while it is not free of **warmup/cooldown bubbles**. As discussed in Appendix E.1, repeating bubbles occur in V-Half only when $T_W < 2\max(T_F, T_B) - 2 \min(T_F, T_B)$.
>
> Generally, longer lifespan means looser dependency chain, making the schedule more robust to the variation of running times, and more friendly to overlap communication and computation. That's the intuitive reason why V-Min is vulnerable (minimal lifespan means tightest dependencies) and V-Half is more resilient. While at the memory budget of V-ZB, it is free of both types of bubbles.
>
> >How is the number of stages in the pipeline determined? ... what is the number of stages used in this work? How does the number of stages influence performance?
>
> For a **balanced computation load**, each device is expected to host equal number of stages. That's why strategies like interleaved-1F1B use a stage number which is a multiple of the device number.
>
> For a **balanced peak memory**, we need the total lifespan to be equal on each device. As shown in our paper, it can be achieved using the "V-Shape" schedule, where the number of stages is twice that of the device. Repeating the "V-Shape" into a "W-Shape" would double this ratio, and would be memory balanced as well, but it does not further decrease the memory, and on the throughput it improves at the rate of $\frac{2+V}{V}2d$ (for V-Min) where $V$ is the number of stages per device, which is not very significant. Therefore we didn't bring this extra complexity to readers in the paper.
>
> >Given the constraints of memory ..., do the authors use grid search to find an optimal schedule? Is it possible to ... quickly find a good pipeline schedule?
>
> We elaborate a simple search approach to quickly find the optimal schedule given the memory constraint (the number of possible building blocks is $O(1)$) in Section 3.1, and introduce an adaptive scheduler with finer control granularity (the number of possible building blocks is $O(d)$). We also evaluate its trade-off between memory and bubble in Figure 9 and 10 (Appendix B).

---

> ### Comment · Reviewer_vShK · 2024-08-13
>
> Thank you for the rebuttal and certain rewrites. I will keep my score.

---

### Author Rebuttal · Authors · 2024-08-06

Thanks for all reviewers for the valuable feedback and insightful suggestions. We updated our PDF accordingly. The changes mainly include:
- We change the title from "Efficient Pipeline Parallelism with Controllable Memory" to "Pipeline Parallelism with Controllable Memory.
- We've corrected the MFU numbers in the experiment tables and graphs. We initially calculated the peak FLOPs of A100 as 312 * 1024^4, which should be 312 * 1000^4 instead. This adjustment increases all MFU numbers by about 10%. This adjustment doesn't affect the conclusions or acceleration ratios, as it is applied equally to both the baseline and our methods. In the grid search experiment, our methods can reach 66.34% MFU (~206 TFLOPS/s) on 40 A100s.
- We rewrite Section 3.1 to include more technical details on the control of peak memory for V-Shape schedules. Specifically, we use uniform offsets between adjacent F/B passes across different devices (refer to the PDF), and leave those within the same devices flexible and use brute force to find optimal solutions. Notably, all of V-Min, V-Half and V-ZB result from systematic searches. Not designed by heuristics.
- We improve the presentation by reorganizing the placement of the figures, changing pipeline schedules from 5 devices 10 microbatches to 4 devices 8 microbatches, and adding more details in the captions. (refer to the PDF)


---------------------------------------------------------------------------
# Section 3.1

We assume the model is uniformly partitioned, namely, both the computation and memory of each stage are identical. For a single microbatch, we denote the activation memory of each stage as $m$, and the total activation memory of the entire model as $M$. Note that $M=2dm$, where $d$ is the number of devices. To make it simple and tractable, we use **uniform offsets within each half of F and B passes** to control the peak memory. Specifically, we apply the same offset $\delta_F^0$ between two adjacent F passes within the first $d$ stages (e.g., $\delta_F^0=2$ in Figure 3b, $\delta_F^0=1$ in Figure 3c and $\delta_F^0=4$ in Figure 3d). Similar constraints are applied to the other half of the F passes and both halves of the B passes, denoted as $\delta_F^1, \delta_B^0, \delta_B^1$, respectively. To guarantee balanced peak memory across devices, we add another two constraints, $\delta_F^0=\delta_B^1=\delta^0$ and $\delta_F^1=\delta_B^0=\delta^1$, where we use notations $\delta^0$ and $\delta^1$ for simplicity. For example, in Figure 3d, we set $\delta^0=4$ and $\delta^1=2$. Note that we only control the offsets across different devices. For those adjacent passes within the same device (e.g., F and B of the last stage, two F and two B in the last device), we use brute force to find optimal solutions, ensuring their offsets are small (less than the repeating interval). Note that W can always be placed greedily after settling all F and B passes, so we don't need to search their offsets during brute force.
According to Equation 1, we can analyze the asymptotic peak memory with respect to $d$,
$$\text{peak memory of device }i \leq \frac{2d(\delta^0 + \delta^1) + O(1)}{6} m  \approx \frac{\delta^0 + \delta^1}{6} M $$

By ignoring the small constant, we can directly control the peak memory by the value of $\delta^0$ and $\delta^1$.

By varying the values of $\delta^0$ and $\delta^1$, we come up with 3 novel V-Shape building blocks (Figure 3), and present their final schedules based on our framework in Figure 4.

---

### Decision · Program_Chairs · 2024-09-25

**Decision:**

Accept (poster)

**Comment:**

All reviewers have given favorable ratings for this paper, ranging from 5 to 7.
Average Rating: 6.33 (Min: 5, Max: 7)
The following strengths of the paper have been highlighted. However, as there are few strong recommendations for this paper, an acceptance as a poster presentation is considered appropriate.

1: This paper proposes a novel framework that decomposes pipeline parallelism schedules into memory-efficient building blocks. This approach addresses inefficiencies in existing schedules, designing new schedules that reduce memory usage while improving throughput.

2: The proposed method demonstrates a 7% to 55% improvement in throughput in pure pipeline parallelism settings compared to 1F1B. Additionally, it achieves a 16% throughput improvement over the 1F1B baseline for large language models.

3: The paper presents a systematic methodology focused on balancing memory usage and reducing pipeline bubbles. The evaluations confirm the effectiveness of the proposed method through both theoretical and experimental validation.

4: The paper introduces new V-Shape building blocks, such as V-Min, V-Half, and V-ZB, which reduce memory usage by more than half compared to traditional methods while maintaining comparable throughput.